# Development of Loop-Mediated Isothermal Amplification (LAMP) Assays for the Rapid Detection of Toxigenic *Aspergillus flavus* and *A. carbonarius* in Nuts

**DOI:** 10.3390/ijms25073809

**Published:** 2024-03-29

**Authors:** Wanissa Mellikeche, Alessandra Ricelli, Giulia Casini, Marilita Gallo, Nuray Baser, Giancarlo Colelli, Anna Maria D’Onghia

**Affiliations:** 1Department of Agricultural Sciences, Food, Natural Resources and Engineering, University of Foggia, Via Napoli, 25-71122 Foggia, Italy; wanissa.mellikeche@unifg.it (W.M.); giancarlo.colelli@unifg.it (G.C.); 2National Research Council—Institute of Molecular Biology and Pathology, P.le A. Moro, 5-00185 Rome, Italy; 3Enbiotech SRL, Via Del Bersagliere, 45-90143 Palermo, Italy; g.casini@enbiotech.eu; 4International Centre for Advanced Mediterranean Agronomic Studies, Via Ceglie, 9-70010 Valenzano, Italy; gallo@iamb.it (M.G.); baser@iamb.it (N.B.); donghia@iamb.it (A.M.D.)

**Keywords:** LAMP, *Aspergillus*, pistachio, almond, mycotoxins, postharvest

## Abstract

*Aspergillus* species create major postharvest problems due to the food losses caused by their mere presence and the hazardous mycotoxins they produce, such as aflatoxin B1 (AFB1) and ochratoxin A (OTA). These mycotoxins are mainly produced by *A. flavus* and *A. carbonarius*, respectively. In this study, we developed a rapid detection method for the two aforementioned species based on loop-mediated isothermal amplification (LAMP). The primers were designed to target genes belonging to the mycotoxin clusters *pks* and *aflT* for *A. carbonarius* and *A. flavus*, respectively. Result visualization was carried out in real time via the detection of fluorescent signals. The method developed showed high sensitivity and specificity, with detection limits of 0.3 and 0.03 pg/reaction of purified DNA of *A. carbonarius* and *A. flavus*, respectively. The assays were further implemented on inoculated nuts, including pistachios and almonds, after one-step crude DNA extraction. These tests revealed a detection level of 0.5 spore/g that shows the effectiveness of LAMP as a rapid method for detecting potentially toxigenic *Aspergillus* spp. directly in food. The validation of the assays included tests on a larger scale that further confirmed their sensitivity and specificity, as well as enabling the production of ready-to-use LAMP prototype kits. These kits are easy to use and aim to simplify the screening of food samples in order to monitor the presence of specific *Aspergillus* contaminations.

## 1. Introduction

Tree nuts have always been produced in specific regions where the climatic conditions meet their particular needs to grow and develop. In these regions, nuts are not considered mere snacks, but important human dietary components providing energy, fatty acids, antioxidants and minerals for local populations. Their high nutritional value has resulted in increasing consumer demand worldwide, to which the major cultivating nations have responded with intensified production and exportation. Therefore, nuts have become critical economic assets for those countries, and their proliferation has increased the need for long-distance transportation and longer storage periods. However, while these developments have facilitated greater human consumption of beneficial nutritional compounds, they can result in the formation of toxigenic *Aspergillus* molds. Moreover, the postharvest handling required for nuts often enhances the distribution of these pathogens, whose mere presence can cause spoilage, but are particularly harmful due to their mycotoxin production.

Mycotoxins are a group of fungal secondary metabolites that includes some of the most hazardous natural substances to human health. Several molds are able to produce and release them in and on food, making it unfit for consumption [1]. Among the most dangerous mycotoxins are ochratoxin A (OTA) and aflatoxin B1 (AFB1). OTA is a nephrotoxic, carcinogenic, teratogenic, immunotoxic and hepatotoxic mycotoxin [2]. Furthermore, it is cytotoxic to the intestinal epithelium and the mucosa-associated lymphoid tissue [3]. In mild and tropical climates, it is produced mainly by black *Aspergilli* (section *Nigri*) such as *A. niger* and *A. carbonarius* [4]. In particular, *A. carbonarius* is known for its strain-related high OTA production levels. Aflatoxins are related to species belonging to green *Aspergilli* (section *Flavi*). There are four main aflatoxins (B1, B2, G1 and G2), all of which pose concerns for food safety due to their effects on the liver and interference with the immune system [5]. However, aflatoxin B1 (AFB1) is considered the most hazardous due to its metabolization in the liver by the cytochrome P450 enzyme system (CYP450) as the actively carcinogenic aflatoxin B1-8-9-epoxide [6].

After the discovery of AFB1 and OTA in 1960 and 1965, respectively, and the establishment of their high toxicity and their role in food-related outbreaks, several countries issued regulations to control their presence in marketed and imported food. For instance, in Italy, aflatoxins were first regulated in 1965, when a maximum tolerable content of 50 µg/kg in peanuts was set [7]. These rules have since been strengthened and applied to many other susceptible foods. Today, more than 120 countries have set maximum tolerable levels of mycotoxins, each of which differs according to the affected commodity. Nuts are among the most strictly regulated commodities given their ease of contamination by toxigenic fungal pathogens. For instance, the EU has issued several regulations in order to impose maximum tolerable limits of AFB1 and OTA in nuts (Figure 1). Among these nuts, pistachios and almonds, which are often subjected to prolonged commercial storage, are known to be very susceptible to *Aspergillus* molds and the mycotoxins they can synthesize. This is due to their high content of proteins and unsaturated fats, as well as their low amounts of water, which prevent spoilage by other microorganisms [8].

The necessity of respecting these regulations requires the development of effective methods for detecting mycotoxins. Furthermore, although regulations do not target the toxigenic fungal strains, their early and rapid detection may indicate the possible presence of mycotoxins and, moreover, this is fundamental for decision-making. Therefore, PCR assays that target mycotoxin production-related genes have been extensively developed [9,10,11].

Although they are powerful, effective tools for the detection of *Aspergillus*, these assays require numerous steps, including preparatory fungal isolation and purification followed by DNA extraction, clean-up and concentration. These steps are time-consuming and laborious, and require advanced laboratory training. In 2000, Notomi et al. [12] invented a new method, loop-mediated isothermal amplification (LAMP). LAMP is rapid and represents a highly sensitive and specific DNA amplification method. It is based on the use of four to six specific primers and enables detection of the target gene directly from crude DNA extracted from food samples. Several LAMP assays have been established to detect bioremediation processes [13], food pathogens [14] and postharvest contaminants [15]. Amongst these, some focused on toxigenic *Aspergillus* contaminations in foods [11,16,17,18,19]. However, none was developed specifically for pistachios and almonds, despite how prone these nuts are to mycotoxin contamination.

The objective of this study is to design LAMP primer sets targeting genes related to the production of AFB1 and OTA in *A. flavus* and *A. carbonarius*, respectively, and to develop a rapid and simple protocol for the detection of these contaminants in samples of pistachios and almonds. The designed primers and the developed protocol will then be used for the production of user-friendly prototype kits that can be applied on site to detect these contaminants.

## 2. Results

### 2.1. LAMP Primer Design

Using Primer Explorer V5, several sets were designed for both *A. carbonarius* and *A. flavus*; two loop primers (LF and LB) were designed manually for each set to increase the specificity and rapidity of the reaction. The alignment process allowed the verification of primer compatibility with the available target DNA sequences in the NCBI GenBank, while simultaneously excluding DNA sequences from phylogenetically related non-target species to avoid off-target binding. All the resulting sets were tested with the pure DNA of the target and non-target species and those that yielded the best results were selected. These were set as ACS02 for *A. carbonarius* and AFS03 for *A. flavus* (Table 1). The visualization of the real-time LAMP assay results is shown in Figure 2a,b. A positive reaction is represented by a peak of the fluorescence signal, which is independent of the quantity of the DNA.

### 2.2. Specificity and Sensitivity

Although LAMP assays aim to detect crude DNA extracted directly from food samples, they are often tested first on pure DNA to determine their sensitivity level before further implementation on food material. The two LAMP assays developed in this study showed an optimal performance at 63 °C throughout a one-hour reaction. However, they yielded different detection levels of pure target DNA. ACS02 detected 0.1 ng/mL of *A. carbonarius* pure DNA, which is equivalent to 100 pg/mL and 0.3 pg/reaction. Concerning AFS03, the detection limit was 0.01 ng/mL, which equals 10 pg/mL and 0.03 pg/reaction (Figure 3). Furthermore, primary specificity tests showed that the selected primer sets did not detect the tested non-target species (Figure 3).

Once the optimal conditions of the reactions as well as their sensitivity and specificity were determined, mixes containing the primers and the dye were prepared in 0.2 mL disposable tubes. These mixes were dehydrated in order to maintain their stability and concentration. The test of dehydrated primer sets after 15 days of storage at room temperature revealed that they remained stable and maintained the same specificity and sensitivity. In addition to safely storing the primers, this step aims to prepare ready-to-use primers instead of the operator having to prepare the mix for each LAMP run. Therefore, it makes the method easier to apply and more user-friendly.

### 2.3. Evaluation of the LAMP Assays on Food Material

Crude DNA was extracted from artificially inoculated samples and analyzed by the two LAMP assays separately for *A. carbonarius* and *A. flavus*. The LAMP primers were able to detect all the inoculation levels including the 2 spore/sample, which is equivalent to 0.5 spore/g. This is due to the pre-enrichment phase, which allowed spores to germinate and the mycelium to grow for 24 h. The filtration through Whatman paper permitted the elimination of most inhibitors that might have been released by the pistachios during the pre-enrichment, while keeping only the fungal mycelium to be used for the DNA extraction. None of the non-inoculated samples was detected. The test was repeated with samples inoculated with both *A. carbonarius* and *A. flavus* with two repetitions for each. Both assays detected all the tested concentrations including the lowest one of 0.5 spore/g (Figure 4).

### 2.4. Validation

The validation step aimed to establish a comparative study between the LAMP assays and the traditional method of detection of *Aspergillus* spp. It demonstrated that the results obtained with LAMP are reliable and equivalent to the ones obtained by cultural isolation followed by morphological and PCR identification.

The sensitivity test detected no false positive nor false negative results for either the pistachios and almonds. Therefore, the result was 100% for the calculated parameters i.e., relative accuracy, sensitivity of an alternative method and sensitivity of the reference method (Appendix A).

Concerning the RLOD test, the average detection limit was set at 20 spore/sample, which is equivalent to 0.5 spore/g. At this level, no false negative results were obtained. This shows that the LAMP assays for *A. carbonarius* and *A. flavus* are just as sensitive as the reference method (Appendix A).

The specificity of the two LAMP assays was further confirmed by the inclusivity (Figure 5; Appendix A) and exclusivity tests (Figure 6; Appendix A), which resulted in the detection of all the isolates from the target species and no isolates from non-target species in both LAMP assays.

### 2.5. Production of LAMP Kits

Results obtained from the validation confirmed the effectiveness of the two assays; therefore, LAMP detection prototype kits have been produced for *A. carbonarius* and *A. flavus*. These kits are user-friendly and require only basic laboratory training to apply. In addition, they do not require a full laboratory, which facilitates the monitoring of the presence of these fungi on nuts in the field, during postharvest processing and after obtaining the final products.

## 3. Discussion

Routine screening of nuts to detect mycotoxin contamination is crucial to ensure food safety considering their susceptibility to *Aspergillus* species, which produces some of the most strictly regulated mycotoxins. *A. flavus* and *A. carbonarius* are the highest producers of AFB1 and OTA, respectively, mycotoxins whose consumption is associated with several health hazards for humans and animals. The detection of these species with traditional cultural methods or PCR-based molecular methods is time-consuming and laborious and requires expensive laboratory material. Therefore, real-time LAMP has been suggested as an alternative method that maintains the high specificity and sensitivity of PCR while requiring less training, time and material. Several studies that have been conducted to develop LAMP primers targeting *Aspergilli* have produced promising results. For instance, the study of Luo et al. [12] designed LAMP assays that detected 2.4, 7.6 and 20 pg/reaction of pure DNA of *A. flavus*, *A. nomius* and *A. parasiticus*, respectively.

In this study, two real-time LAMP assays (AFS03 and ACS02) were developed to target *pks* and *aflT* genes for *A. carbonarius* and *A. flavus*, respectively. The results were observed in real time by measuring the fluorescence signal emitted by the DNA-binding dye EvaGreen along a 60 min reaction at 63 °C. The assays were first tested on the purified DNA of target and non-target species and showed high specificity and even lower detection levels compared to the above-mentioned study, with a sensitivity of 0.03 pg/reaction and 0.3 pg/reaction for *A. flavus* and *A. carbonarius*, respectively.

LAMP assays are often developed for implementation on susceptible food materials in order to facilitate the detection of target pathogens. Luo et al. [18] developed LAMP assays to detect *A. flavus*, *A. nomius* and *A. parasiticus* on Brazil nuts, maize and peanuts, with detection limits of 10 conidia/g in Brazil nuts, 10^2^ conidia/g in peanuts and 10^4^ conidia/g in maize for *A. flavus*. Furthermore, the study of Niessen et al. [19] aimed to develop LAMP assays targeting the gene *nor1* of aflatoxigenic *Aspergilli* on several foods, including nuts. This test’s detection limit was set at 211 conidia per reaction. In this study, the developed real-time LAMP assays were primarily tested on artificially contaminated pistachio samples. The assays showed their ability to detect relatively very low contamination levels ranging between 0.5 and 2.5 spore/g. This was due to the pre-enrichment step, which enabled a remarkable increase in the assays’ sensitivity by allowing *Aspergillus* spores to germinate over 24 h of incubation.

The LAMP assays’ validation is a further large-scale verification of their efficacy and a critical step towards their commercialization as reliable detection kits for target pathogens. This is based on applying a series of tests to real-time LAMP assays while referring to commonly used methods (cultural and PCR). It allows determination of the percentages of false negatives and false positives, evaluation of the limit of detection and study of the specificity and sensitivity of the sets on a larger scale. Previous validation studies on real-time LAMP showed how reliable it is as an alternative method to detect food contaminants compared to PCR [20]. In our study, the validation step further confirmed the detection level of the two assays at 0.5 spore/g while ensuring a very low probability of obtaining false negative or false positive results. Furthermore, the inclusivity and exclusivity tests showed the high specificity of the assay. However, it would be interesting for future studies to also test the efficacy of the primers on *A. welwitschiae* and *A. nomius*, which are phylogenetically related to *A. carbonarius* and *A. flavus*, respectively.

## 4. Material and Methods

### 4.1. Fungal Strains

Fungal strains used for the development, optimization and validation of LAMP primer sets were obtained from the CIHEAM collection, Bari, Italy (Appendix A). The strains of *A. carbonarius* and *A. flavus* were previously confirmed to be OTA and AFB1 producers, respectively, by HPLC analysis. Other strains of *A. parasiticus*, *A. tamarii*, *A. niger*, *A. tubingensis* and *Penicillium* spp. were selected as non-target species to test the primers’ selectivity.

### 4.2. DNA Extraction

Mycelium of each target and non-target isolate was grown in 50 mL of PDB for four days at 25 °C in a shaking incubator (120 rpm). The mycelium was then collected by filtration and DNA was extracted according to the method of Carlucci et al. [21]. The concentrations of extracted DNA were determined by a spectrophotometer, and aliquots of equal concentrations were prepared and used for primer optimization tests.

### 4.3. Crude DNA Extraction

Crude DNA was extracted directly from 4 g of inoculated pistachio nuts after a 24 h pre-enrichment step in 36 mL of PDB at 25 °C. The liquid medium was then filtered through Whatman paper using a filtration ramp (WaterVac 100, Rocker, Kaohsiung, Taiwan) to collect the germinated spores. The filter was then incubated in 500 µL of a one-step extraction buffer (Enbiotech Srl, Palermo, Italy) at 95 °C for 10 min. During the validation step, the same protocol was followed for the extraction of crude DNA from fungal mycelium in pistachio and almond samples. However, to ensure accurate representation, larger samples (20 g) were pre-enriched in 180 mL of PDB of which 40 mL were filtered and used for DNA extraction.

### 4.4. Primer Design

LAMP primer sets were designed to target suitable specific genes within aflatoxin and ochratoxin gene clusters as follows: *aflT* for *A. flavus* (Figure 7) and *pks* for *A. carbonarius* (Figure 8). The *pks* gene has been proven to be involved in the biosynthesis of ochratoxin A by *A. carbonarius* strains [22]. Although *aflT* does not seem to be directly related to the aflatoxin biosynthesis pathway, it is part of the AFB1 cluster and codes for a fungal transporter among the major facilitator superfamily [23]. Furthermore, this gene allowed the design of species-specific primers to detect *A. flavus.* Gene sequences for target species and phylogenetically close microorganisms were obtained from the NCBI gene database https://www.ncbi.nlm.nih.gov/gene/ (accessed on 31 January 2022). The alignment of these genes was conducted using the Clustal Omega tool https://www.ebi.ac.uk/Tools/msa/clustalo/ (accessed on 10 February 2022) in order to include all available strains and exclude closely related species. LAMP primer sets were designed using primer explorer 5 provided by Eiken Chemical Co., Ltd. Tokyo, Japan http://primerexplorer.jp/lampv5e/index.html (accessed on 10 February 2022) in. Each set contained two inner primers (FIP and BIP), two outer primers (F3 and B3) and two loop primers (LF and LB). Loop primers were manually designed to expedite the reaction and increase its sensitivity.

### 4.5. LAMP Reaction and Result Visualization

The LAMP assay was carried out using the ICGENE, a LAMP-specific device provided by Enbiotech Srl and linked via Bluetooth to a smart tablet equipped with the pre-installed ICGENE application. The device is constituted of four blocks and each block can hold up to 12 samples. This setting allows real-time detection and automatic interpretation of results. Primers, DNA and double strand intercalating DNA-binding dye EvaGreen (Biotium, Inc., Fremont, CA, USA) were added to a LAMP mix (Enbiotech Srl) containing a strand-displacing DNA polymerase, dNTPs and salts needed for the amplification. Results were obtained by measuring the increase in fluorescence signal throughout one hour and are represented by one graph per block (12 samples). For each LAMP run, purified DNA was used as a positive control and nuclease-free water as a negative control.

### 4.6. Optimization of the LAMP Assay Conditions

LAMP primer sets were tested on purified *A. carbonarius* BP36 and *A. flavus* BP53 DNA, and those that gave promising results were subjected to the process of optimization. This was achieved by adjusting the primer concentrations and the proportions of the other components of the reaction (e.g., LAMP mix and quantity of DNA). The reaction was repeated at various temperatures (55 °C, 60 °C and 63 °C) to determine the optimal conditions for enhancing the efficacy of the reaction.

### 4.7. Specificity and Sensitivity of the Primers

The specificity of a detection method represents its ability to accurately detect only its target pathogen while the sensitivity represents its ability to detect it even at low concentrations. Therefore, these two parameters are among the most important targets of a LAMP assay optimization. Specificity was determined by testing purified DNA from non-target species that are phylogenetically close to the targets (Table 1 annex). Sensitivity was determined by testing 10-fold dilutions of purified DNA *of A. carbonarius* BP36 and *A. flavus* BP53.

### 4.8. Primer Dehydration

In order to properly store the LAMP primers while maintaining their stability and functionality as well as to facilitate their use, primer mixes were prepared in 0.2 mL testing tubes and dehydrated using a centrifugal concentrator (Labconco, Kansas City, MO, USA) set at 60 °C for 15 min. The stability of the dehydrated primers was assessed by comparing the results of a first LAMP run immediately after the dehydration and a second run after 15 days of storage at room temperature.

### 4.9. Evaluation of the Primer Sets on Food Material

Intact pistachio nuts were chosen as the matrix on which the LAMP assays were implemented, given their susceptibility to the toxigenic *Aspergillus* spp. However, in order to accurately evaluate the primers, the nuts were first sterilized and then artificially inoculated. The sterilization was carried out by soaking in a 0.5% chlorine sodium hypochlorite solution for 3 min, rinsing thoroughly with water, then drying in the oven at 50 °C. For the inoculation, spore suspensions were prepared by collecting spores from 10-day-old colonies in Petri dishes and preparing 10-fold dilutions, at the concentrations of 1, 10 and 100 spore/mL. One mL was added to 4 g samples of pistachios then the inoculum was stabilized by keeping the samples at 4 °C for 24 h. The inoculation was carried out with *A. flavus* and *A. carbonarius* separately and then together as one inoculum. To enhance the germination of spores, the samples were pre-enriched by adding 36 mL of PDB and incubated at 25 °C for 24 h. Crude DNA was then extracted as previously reported. For each experiment, two controls were considered, non-inoculated pistachios and PDB alone with the inoculum in the absence of pistachios.

### 4.10. Validation of LAMP Assays

In the absence of specific normative guidelines, a validation protocol was established based on ISO 16140:2017 [24] guidelines, which aims to validate an alternative method (in this case LAMP) by comparing it to a reference method commonly used for the pathogen’s detection. However, there is no well-established reference method for the detection and identification of *Aspergillus* spp. Therefore, the reference method was determined based on the literature (Table 2). The validation was performed in three steps: a sensitivity test, a relative level of detection (RLOD) test and a selectivity test on both the pistachios and almonds.

### 4.11. Sensitivity Test

For each method (alternative and reference), each food commodity (pistachios and almonds) and each pathogen (*A. flavus* and *A. carbonarius*), 20 sterilized samples of 20 g each were considered, 10 samples were artificially inoculated at a low contamination level (0.5 spore/g) and 10 were not inoculated. The inoculated nuts were left to stabilize for 24 h at 4 °C. The contaminants were detected from all the samples by the alternative and reference methods, whereby the values shown in Table 3 were determined.

### 4.12. Relative Level of Detection (RLOD) Test

RLOD represents the ratio between the LOD of the alternative method and the reference method, and it is calculated as follows: RLOD = LOD alternative/LOD reference × 100. A total of 30 samples were considered; 5 non-inoculated samples, 20 inoculated at a low level (0.5 spore/g) and 5 inoculated at a high level (5 spore/g). The samples were then analyzed with both methods to determine the RLOD.

### 4.13. Selectivity

This test is divided into two separate ones for inclusivity and exclusivity. Inclusivity is the detection of a range of isolates of the target species with the alternative method. For this test, 10 strains of *A. carbonarius* and 20 strains of *A. flavus* were used. Exclusivity is the detection of non-target isolates with the alternative method. In total, 30 non-target isolates were analyzed with the *A. carbonarius* assay and 15 were analyzed with the *A. flavus* assay.

### 4.14. Production of Prototype LAMP Kits for the Detection of A. flavus and A. carbonarius

The designed primers and the developed protocols for DNA extraction, LAMP amplification and result visualization were then implemented into the production of molecular kits containing all the necessary tools to successfully apply this detection method.

## 5. Conclusions

This study has successfully developed the first user-friendly real-time LAMP prototype prototype kits to detect potentially toxigenic *A. carbonarius* and *A. flavus* directly from nut samples. The prototype kits contain ready-to-use dehydrated primer mixes implemented in a simple protocol suitable to be performed by non-experts. The kits demonstrated high specificity and sensitivity levels and enabled fast and easy result interpretation. These results, compared to previous ones, show the promising and fast progress of real-time LAMP as a replacement for laborious PCR-based amplification methods.

In this study, the two real-time LAMP assays were applied to almond and pistachio nuts; however, they can be slightly modified to fit application to testing any kind of food material. These prototype kits can be a means of ensuring early detection of these contaminant fungi in order to aid future decision-making as well as to expedite and facilitate the screening of marketed samples.

## Figures and Tables

**Figure 1 ijms-25-03809-f001:**
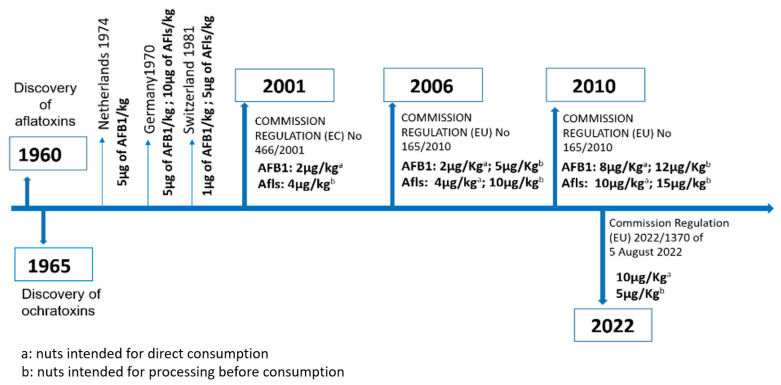
Regulations targeting the maximum accepted level of aflatoxins and OTA in nuts in European countries.

**Figure 2 ijms-25-03809-f002:**
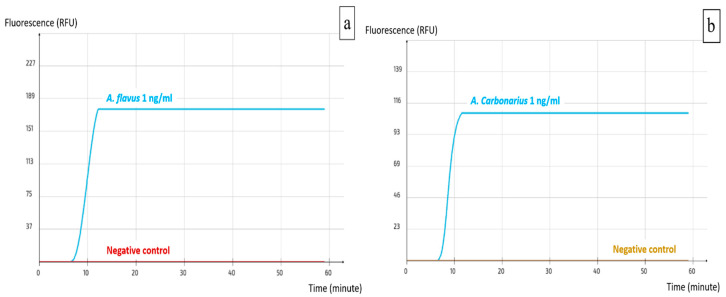
Visualization of real-time LAMP assay results. (**a**) AFS03 for the detection of *A. flavus*; (**b**) ACS02 for the detection of *A. carbonarius*.

**Figure 3 ijms-25-03809-f003:**
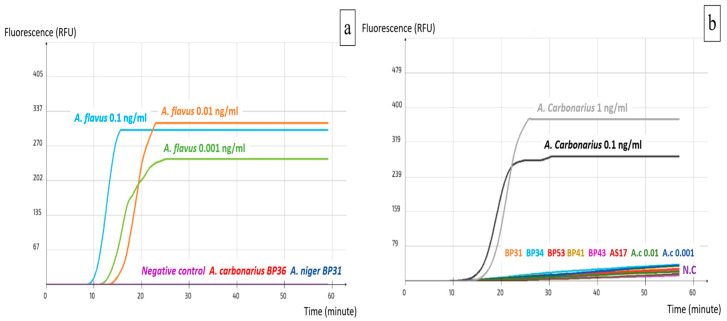
Specificity and sensitivity of LAMP assays: (**a**) AFS03; (**b**) ACS02.

**Figure 4 ijms-25-03809-f004:**
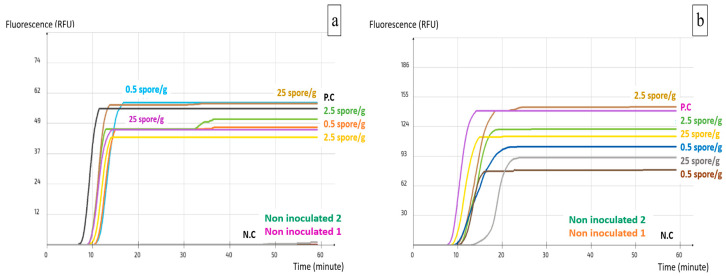
Implementation of the LAMP assays on pistachio nuts: (**a**) AFS03; (**b**) ACS02. P.C: Positive control; N.C: Negative control.

**Figure 5 ijms-25-03809-f005:**
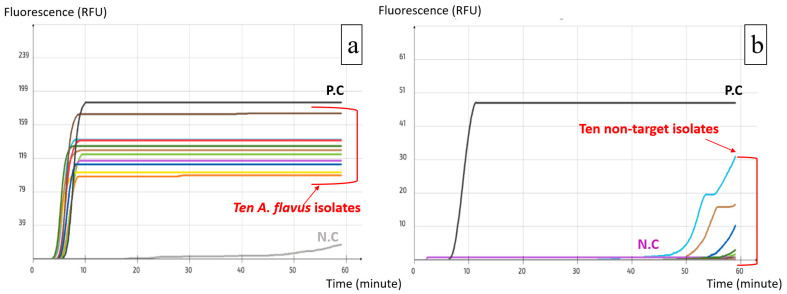
Selectivity test using A. flavus LAMP assay: (**a**) Inclusivity test, successful amplification of *A. flavus* isolates; (**b**) Exclusivity test, failure to amplify non-target isolates. (each colored line corresponds to one target isolate).

**Figure 6 ijms-25-03809-f006:**
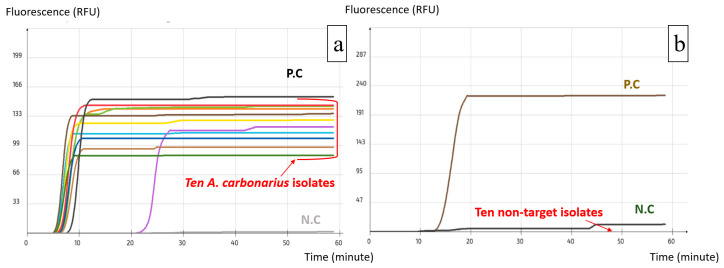
Selectivity test using A. carbonarius LAMP assay: (**a**) Inclusivity test, successful amplification of A. carbonarius isolates; (**b**) Exclusivity test, failure to amplify non-target isolates. (each colored line corresponds to one target isolate).

**Figure 7 ijms-25-03809-f007:**
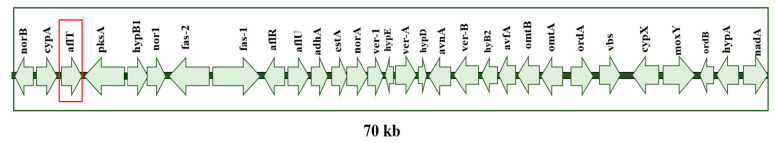
Aflatoxin gene cluster for *A. flavus*.

**Figure 8 ijms-25-03809-f008:**
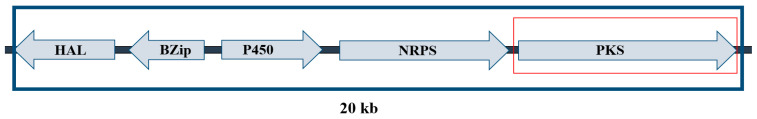
Ochratoxin A gene cluster for *A. carbonarius*.

**Table 1 ijms-25-03809-t001:** Primer sets ACS02 for *A. carbonarius* and AFS03 for *A. flavus*.

Target	Primer Name	Sequence	T (°C)	bp
*A. carbonariuspks*	ACWM02-F3	GCGCAATGCGGCTTCA	55.18	17
	ACWM02-B3	AATAGGACGCGGGTTCTG	56.75	18
	ACWM02-FIP	AGCGTGTGACACAGGTCGTTttttGCATCGCATCTGGAGTCG	62.68/58.38	42
	ACWM02-BIP	GAGAAGAGGACCCGGTTCGATttttAGTGAATTGCGCAGACTGTC	61.3/59.13	45
	ACWM02-LF	CATCCGAGCCCGTCATC	57.16	17
	ACWM02-LB	GCAGCTACCATGTGCCT	56.83	17
*A. flavus* * aflT*	AFWM03-F3	CCAACCGGAGTACATGGAC	59.11	19
	AFWM03-B3	AGACAAAGTTAGCCGGGTAC	58.90	20
	AFWM03-FIP	GCAAATTAGGGCCACCTCAATTTTCAGGTGCTGGTGGCATAT	60.24/59.76	42
	AFWM03-BIP	TCTTAGCATTCTCGGCGCCTTTTCGGTCGTTACCTGTTCTGC	61.76/59.88	42
	AFWM03-LF	TCCAGCAACGCGGCATT	61.95	17
	AFWM03-LB	TCGGGATCGAATGGAGAAGC	61.26	20

**Table 2 ijms-25-03809-t002:** Reference and alternative methods followed for validation tests.

Reference Method	Alternative Method
Isolation of the pathogen from nuts seeded on PDA medium for 48 h at 25 °C Observation of morphological characteristics Pure DNA extraction Molecular identification by PCR using calmodulin gene	Pre-enrichment of 20 g of contaminated nuts Crude DNA extraction Molecular identification by LAMP assay

**Table 3 ijms-25-03809-t003:** Values calculated for the sensitivity test; PA: positive agreement (both methods are positive); NA: negative agreement (both methods are negative); PD: false positive (negative for reference; positive for alternative); ND: false negative (positive for reference; negative for alternative).

Sensitivity of the alternative method	SE_alt_ = (PA + PD)/(PA + ND + PD) × 100%
Sensitivity of the reference method	SE_ref_ = (PA + ND)/(PA + ND + PD) × 100%
Relative accuracy	RT = (PA + NA)/N × 100%
Ratio of false positive results for the alternative method after confirmation	FPR = FP/(PD + NA) × 100%

## Data Availability

The raw data supporting the conclusions of this article will be made available by the authors on request.

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
