# Peer review of "Development of Loop-Mediated Isothermal Amplification (LAMP) Assays for the Rapid Detection of Toxigenic Aspergillus flavus and A. carbonarius in Nuts"

_ijms, 2024, doi:10.3390/ijms25073809_

Round 1
Reviewer 1 Report
Comments and Suggestions for Authors
General
The paper is mostly well-written and scientifically sound. I would like to see a clear statement of how the present work improves upon or is different than the work of Luo et al – is the present work simply on a different matrix?
Several points in the text mention development of a ‘kit’. One author is affiliated with Enbiotech and the work presented here makes use of Enbiotech supplies and equipement. A reader might conclude that the work presented in this manuscript is for the development of a product that will be commercialized by one of the author’s employers. Is this not a conflict of interest?
I know that the journal suggests placing the results and discussion before the materials and methods, but the results only make sense after reading the methods. This is especially true with a paper of this type which is the development of a method. I would encourage the authors to format the paper with the methods before the results and discussion.
Specific:
Line 77-78: The text up to this point has explained the problem of mycotoxin contamination and regulatory framework. The sentence here, however, seems misleading. The regulations do NOT require detection of the fungus itself, nor is there any legal prohibition against the presence of Aspergillus in commodities. The regulations are only applicable to the toxins. We can agree that the presence of Aspergillus is often linked to the presence of aflatoxin and that there is value in the detection of the fungus, but it is not a regulatory requirement.
Line 94-95: Can you clarify this point – Of the 5 LOOP papers you cite here did any of them test pistachios and almonds?
Figure 3: I would suggest a larger presentation of this figure, or if it is the same size, at least increase the size of the text within the figure.
Line 152: Write out “spore” instead of just “s”
Figure 4: I would suggest a larger presentation of this figure, or if it is the same size, at least increase the size of the text within the figure. Write out “spore” instead of just “s”
Figure 3: I expected to see a figure showing amplification of A. flavus and A. carbonarius AND the failure to amplify all the other tested species
Line 222: What is this collection?
Line 234: Are these intact pistachios or cracked or ground? How were they inoculated and for how long? Is this the same inoculation as described in section 4.9? If so, move that section ahead of this section.
Line 236: What is a filtration ramp?
Line 290: What are “internal standardized conditions”?
Line 336: You say “Table 1” but I think you mean the supplemental Table 1
Line 343: The text here and elsewhere regarding the production of ‘kits’ suggests a deployment and/or commercialization of this method. Are these kits in use? Are they available for others?
Author Response
Please see the attachement. Thank you.

Reviewer 2 Report
Comments and Suggestions for Authors
The topic is very relevant, the objective of this study being to design LAMP primer sets targeting genes related to the production of AFB1 and OTA in A. flavus and A. carbonarius, and to develop a rapid and simple protocol for the detection of these contaminants in samples of pistachios and almonds.
The methodology is very modern, using molecular biology and bioinformatics tools.
The results are very important for detection of mycotoxins in food and feed. This study has successfully developed the first user-friendly real-time LAMP kits available on the market to detect potentially toxigenic A. carbonarius and A. flavus directly from nut samples. The kits demonstrated high specificity and sensitivity levels and enabled fast and easy result interpretation.
The conclusions are consistent with the evidence and arguments presented.
The references are very relevant, including also some relevant author’s previous experience in the field.
I suggest some minor editing corrections
1. Citation of references in text should be made according to MDPI rules (numbers of ref, instead of names). Line 86 In 2000, Notomi et al (17) invented a new method
2. Number of Tables must be put in order of the appearance in text
3. In Introduction, when you mentionthe toxic effect of ochratoxin, you may see also
C. Solcan, G. Pavel, VC Floristean, SI Beschea Chiriac, BG Şlencu, G Solcan, 2015, Effect of ochratoxin A on the intestinal mucosa and mucosa-associated lymphoid tissues in broiler chickens, Acta Veterinaria Hungarica vol. 63, 1, pp. 30–48, 2015, DOI: 10.1556/AVet.2015.004
